# Assessing suicidality during the SARS-CoV-2 pandemic: Lessons learned from adaptation and implementation of a telephone-based suicide risk assessment and response protocol in Malawi

Kelsey R. Landrum[1]*, Christopher F. Akiba[2], Brian W. Pence[1], Harriet Akello[3], Hamis Chikalimba[3], Josée M. Dussault[1], Mina C. Hosseinipour[4], Kingsley Kanzoole[5], Kazione Kulisewa[6], Jullita Kenala Malava[3], Michael Udedi[7], Chifundo C. Zimba[3], Bradley N. Gaynes[1,8]

1 Department of Epidemiology, Gillings School of Global Public Health, University of North Carolina at Chapel Hill, Chapel Hill, North Carolina, United States of America, 2 Department of Health Behavior, Gillings School of Global Public Health, University of North Carolina at Chapel Hill, Chapel Hill, North Carolina, United States of America, 3 UNC Project Malawi, Lilongwe, Malawi, 4 University of North Carolina at Chapel Hill, Chapel Hill, North Carolina, United States of America, 5 Abwenzi Paza Umoyo/Partners in Health, Neno, Malawi, 6 Department of Psychiatry and Mental Health, Kamuzu University of Health, Blantyre, Malawi, 7 Noncommunicable Disease and Mental Health Unit, Malawi Ministry of Health, Lilongwe, Malawi, 8 Department of Psychiatry, University of North Carolina at Chapel Hill, Chapel Hill, North Carolina, United States of America

* klandrum@email.unc.edu

**Data Availability Statement:** The data repository is located at https://nda.nih.gov/edit_collection.html?

## Abstract

The SARS-CoV-2 pandemic led to the rapid transition of many research studies from in-person to telephone follow-up globally. For mental health research in low-income settings, tele-follow-up raises unique safety concerns due to the potential of identifying suicide risk in participants who cannot be immediately referred to in-person care. We developed and iteratively adapted a telephone-delivered protocol designed to follow a positive suicide risk assessment (SRA) screening. We describe the development and implementation of this SRA protocol during follow-up of a cohort of adults with depression in Malawi enrolled in the Sub-Saharan Africa Regional Partnership for Mental Health Capacity Building (SHARP) randomized control trial during the COVID-19 era. We assess protocol feasibility and performance, describe challenges and lessons learned during protocol development, and discuss how this protocol may function as a model for use in other settings. Transition from in-person to telephone SRAs was feasible and identified participants with suicidal ideation (SI). Follow-up protocol monitoring indicated a 100% resolution rate of SI in cases following the SRA during this period, indicating that this was an effective strategy for monitoring SI virtually. Over 2% of participants monitored by phone screened positive for SI in the first six months of protocol implementation. Most were passive risk (73%). There were no suicides or suicide attempts during the study period. Barriers to implementation included use of a contact person for participants without personal phones, intermittent network problems, and pre-paid phone plans delaying follow-up. Delays in follow-up due to challenges with reaching contact

id=2822. All data used in this analysis are available free of cost to researchers.

**Funding:** This study received funding from the National Institute of Mental Health (PI: Brian W. Pence, PhD; U19MH113202-01) (https://www.nimh.nih.gov). KRL received support from supplement U19MH113202-04S1 (https://www.nimh.nih.gov). JD received support from supplement U19MH113202-01S2 and T32AI070114 (https://www.nimh.nih.gov; https://sph.unc.edu/epid/epid-student-central/epid-student-funding/). Neither the funding agency nor funders had any part in data collection, data analysis, or decision-making about publication.

**Competing interests:** The authors have declared that no competing interests exist.

persons, intermittent network problems, and pre-paid phone plans should be considered in future adaptations. Future directions include validation studies for use of this protocol in its existing context. This protocol was successful at identifying suicide risk levels and providing research assistants and participants with structured follow-up and referral plans. The protocol can serve as a model for virtual SRA development and is currently being adapted for use in other contexts.

## Introduction

Mental health disorders are among the leading causes of death and disability worldwide, sespecially in low- and middle-income countries (LMICs). Globally, mental health disorders account for nearly a third of years lived with disability and are the fifth-leading cause of disability-adjusted life years (DALYs), making up 13% of DALYs worldwide [1–3]. Aproximately three quarters of this burden resides in LMICs [4]. In addition to morbidity and disability, untreated mental health disorders are associated with early mortality with an estimated 75% of suicides occurring in LMICs [5]. Yearly prevalence of suicide attempt and suicidal behavior in Malawi is approximately 0.8% and 7.9%, respectively [6]. While effective and low cost treatments are available for the most common mental disorders, the treatment gap is yawning [7, 8]. The proportion of mental health workers in LMICs is as low as 2 per 100,000 population; accordingly, most affected individuals in LMICs do not receive care [9]. The average disparity between those with mental illness in need of care and those who receive services is over 90% in most LMICs [10–14]. The extremely limited access to and availability of national or local suicide hotlines in this context further indicated need for a telephone-based Suicide Risk Assessment (SRA) during the SARS-CoV-2.

The limited availability of mental health services makes identification and management of suicidal thoughts and behavior, often referred to as suicidality, especially challenging. Such behavior can include passive and active suicidal ideation (SI), suicide attempts, and death due to suicide [15, 16]. Passive SI is defined as having thoughts of suicide without the intention to act on suicidal thoughts, while active SI is defined as having suicidal thoughts and intention to act on such thoughts [15, 17]. Passive and active SI typically require different clinical interventions. Conducting mental health research often requires SI measurement, with careful assessment and triage, and is traditionally dependent on in-person assessment with the ability to immediately engage participants if clinical safety measures are needed [18–27].

The SARS-CoV-2 pandemic disrupted and further limited mental health care by precluding many of these face-to-face assessments [28]. The pandemic forced ongoing studies of mental heath in LMICs to quickly pivot from in-person to virtual SI assessments with consideration of how to safely triage and respond to suicide risk by telephone. Increased suicidal risk has previously been associated with epidemics, with ongoing studies suggesting the possibility of increased risk of suicide during the current SARS-CoV-2 pandemic [29–32]. Further, assessment of the feasibility of adapting in-person mental health assessments to virtual assessments in low-resource settings during the pandemic is essential and missing from scientific literature.

In the midst of an ongoing clinical trial integrating depression treatment in non-communicable disease (NCD) settings in Malawi, researchers and participants in the Sub-Saharan Africa Regional Partnership for Mental Health Capacity Building (SHARP) study underwent a rapid change from in-person suicidality assessments to telephone-based assessments by developing a

protocol to provide a robust, feasible, and rapidly adaptable protocol to assess suicidality in a timely, safe, quality, affordable, and virtual manner [33–35].

This paper describes the feasibility of development and implementation of this suicidality assessment protocol. We aim to: 1) assess the feasibility (defined as the successful use of the tool in this study's context) of the protocol for assessing suicidality over telephone among patients who screen positive during the Suicide Risk Assessment (SRA), 2) describe the information we were able to collect and categorize, 3) describe the primary challenges identified and lessons learned during protocol development, and 4) discuss future developments and protocol use in other settings [36, 37].

## Methods

The parent, SHARP scale-up study is an implementation science trial comparing the success of a basic versus an enhanced implementation package in achieving integration of depression treatment with NCD treatment at 10 NCD clinics in Malawi [33, 35]. Clinics in the parent study were randomized with a 1:1 ratio to a basic or enhanced implementation strategy to compare implementation and effectiveness outcomes and cost-effectiveness of the two strategies. Patients receiving care at each clinic who meet eligibility criteria described below were invited to enroll in the study to provide clinical outcome measures. In this paper, we describe the feasibility, development, and implementation of the SHARP Safety Response Protocol for Phone Interviews in response to the SARS-CoV-2 pandemic. We do not report results of the parent study trial.

### Participants

The cohort of patients (n = 946) consisted of participants meeting the following parent study inclusion criteria: aged 18–65 years and being a patient in care for diabetes or hypertension management at a participating NCD clinic. Of these patients, n = 739 had elevated depressive symptoms (a score ≥5 on the Patient Health Questionnaire-9 [PHQ-9]) at baseline and were eligible for inclusion in this analysis. Exclusion criteria are history of bipolar or psychotic disorder and/or an emergent self-harm threat. One participant was excluded from analysis due to being a prevalent SI case already receiving regular SI follow-up prior to the study period. Among the n = 738 participants eligible for analysis, n = 602 SI screenings were conducted during the study period.

### Ethical approval

All study materials and research activities have been approved by the National Health Sciences Research Committee of Malawi (NHSRC; Approval # 17–3110) and the Biomedical Institutional Review Board of the University of North Carolina at Chapel Hill (Approval #17/11/1925). All methods were performed in accordance with the relevant guidelines and regulations. The current manuscript does not report results of the parent study. All participants gave written, informed consent to participate in the parent study. Written informed consent clearly explained the purpose of the research, what will be required of the individual, and the risks and benefits of participation in the participant's preferred language. All informed consent procedures for the parent study were approved by both ethics committees.

### Measures: PHQ-9 and SRA

Participants are asked to complete the PHQ-9, a depression screening tool developed and originally validated in high-income countries but subsequently validated in Malawi and many

other low-income countries, as part of baseline and follow-up assessments at 3, 6, and 12 months [38–40]. The final PHQ-9 question asks, "During the past two weeks, how many days have you been bothered by thoughts that you would be better off dead or of hurting yourself in some way?" Participants respond with: *"0 days"*, *"1–7 days"*, *8–12 days"*, or *"13 or 14 days"*. Any response other than *"0 days"* was considered a positive screen for suicidal ideation, requiring a follow-up SRA following this protocol's schedule. The PHQ-9 questionnaire and SRA were translated into Chichewa and Chitumbuka for use in Malawi [41].

## Setting

From May 13, 2019 through March 24, 2020, interview data were collected with in-person interviews at 10 NCD clinics in all three regions of Malawi. However, as a result of the SARS-CoV-2 pandemic and the patient population's high risk for severe complications of COVID-19, all in-person participant contact was put on hold from March 25, 2020 until October 16, 2020. The research team decided to transition to telephone-based follow-up interview data collection, necessitating the development of virtual study and safety assessment and response protocols.

## Protocol development

The SRA protocol guides research team members through a series of risk assessments and the creation of a follow-up plan. The protocol was originally designed to be implemented in-person immediately after a participant screens positive on PHQ-9 Question 9 (any response other than *"0 days"*) or indicates concerns of suicidal thoughts. The tablet-based data collection software, Open Data Kit (ODK), is set to warn the RA to conduct the SRA when the participant answers affirmatively to PHQ-9 Question 9 [42]. If the patient screens positive, the RA is instructed to not leave the patient alone, complete the telephone protocol form, transfer the patient to clinician care, enter the event into the study log for review by team members, and follow up with the participant as indicated by the protocol (SF 1, SF 2). If the participant indicates SI via a method other than the PHQ-9, RA's also completed the SRA.

The research team met in March 2020 to discuss methods of adapting the in-person protocol to assess and manage suicide risk virtually during, and beyond, the SARS-CoV-2 pandemic. Given the immediate need of a suicidality protocol without time for a full-validation study of the protocol before implementation due to the pandemic, subject matter experts helped develop the guide. Anticipating that hospitals would have increased resource strain, decreased patient capacity, and increased SARS-CoV-2 transmission risk, the team revised the SRA for telephone delivery with branching logic to guide trained research assistants (RAs) in creating a follow-up plan with participants, reducing participants' immediate suicide risk, engaging a nearby support person, creating a follow-up plan with participants, and referring participants to in-person care when clinically indicated. Research assistants, with a minimum high school education level and training in SHARP study procedures, were trained by study staff and subject matter experts in suicidality and related health protocols and interventions, including psychiatry, health behavior, and epidemiology experts.

The in-person SRA is an algorithmic questionnaire that is administered by phone in Chichewa or Chitumbuka, and responses are translated into English for data entry. Research staff provide feedback about the event to the research coordinator after each time the protocol is administered. The protocol was implemented in April 2020 after an iterative review process including all research team members with diverse areas of subject matter expertise.

## Procedure: Protocol training

The RAs received two trainings for telephone SRA implementation from clinical and research study team members. These trainings consisted of two 1-hour formal trainings via Zoom in which team members discussed the protocol and practiced protocol implementation in simulated scenarios. Team members had a follow-up meeting one week later to answer questions that arose while preparing for protocol implementation after the two training sessions.

## Procedure: Protocol implementation of initial assessment

**Section 1: Suicide risk assessment (SF 1–2).**   The aim of Section 1 is to assess level of SI. Part A (SF 1) assesses active versus passive suicidality and Part B assesses severity of active thoughts, if present. The RA assesses the participant's current self-harm risk in Part C (SF 1), regardless of the participant's reporting of active versus passive suicidal thoughts in Part A. Suicide risk level is defined based on participant responses to Parts 1A-C. Importantly, "thoughts of hurting yourself right now" (SF 1, C.1) was added to the telephone protocol, as no clinician (who would have asked this question in-person and provided immediate intervention in the pre-COVID period) was available with this virtual assessment. The addition of this question to the telephone protocol allowed immediate, virtual assessment of active high and acute risk status.

**Section 2: Confirm patient location.**   The purpose of Section 2 (SF 1) is to confirm the patient's location and identify if other individuals are nearby to help the participant as needed. The RA continues to Section 3 if the participant's suicide risk is active-high or active-emergent. The RA skips to Section 4 if the participant's suicide risk is passive, active-low, or active-moderate.

**Section 3: Safety plan and next steps for active-high or active-emergent.**   Section 3 (SF 1) is critical in confirming contact information, on-site support persons, and a strategy to assist the patient in reaching mental health care. The RA confirms the participant's emergency contact and Health Surveillance Assistant (HSA) information. An HSA is a trained community-based health provider who is able to link community members to various health services. The RA asks the participant to go to the nearest health facility to speak with a mental health professional with the help of a nearby trusted contact. If no trusted contact is available, the RA speaks with an emergency contact in the participant's locator form regarding the immediate safety concern. If no emergency contact is available, the RA assists the participant in contacting a health professional at the nearest health facility and notifies the study's clinical coordinator at the participant's health facility.

**Section 4: Safety plan and next steps for passive, active-low, or active-moderate.**   The overall goal of Section 4 (SF 1) is to create a safety and follow-up plan. The RA asks the participant to inform a trusted contact if the participant feels as though they might harm themselves. If the participant does not agree, the RA returns to Section 3 to proceed with the active-high and active-emergent workflow, including asking the participant if they can speak with a nearby individual. If the participant agrees to inform a trusted contact if the participant feels as though they might harm themselves, the RA continues through Section 4 to assess the presence of potentially dangerous items in the home. If the RA cannot confirm that the participant has handed over potentially dangerous items, the RA completes Section 3. Otherwise, the RA provides the participant with referral information to mental health specialists and asks the participant to seek care if they believe they may harm themselves.

The protocol then defines a follow-up schedule based on suicide risk level (SF 1–5). If the RA cannot reach the participant at the follow-up time, the RA calls the participant's contact

person. If the RA is unable to reach the participant's contact, the RA informs the study coordinator for further guidance.

## Procedure: Protocol implementation of Section 5: Saftey follow-up and Section 6: Summary of follow-up contacts

During the scheduled follow-up contacts, the RA asks how the participant is doing, for updates on the agreed upon action plan from the previous call, and screens for SI with PHQ-9 Question 9 (SF 1). If no SI is present, the RA continues with the prescribed follow-up schedule. If suicidal risk is present, the RA restarts the protocol, returning to Section 1.

**Passive risk follow-up (SF 1, SF 3).** Follow-up ends if the participant begins with a passive risk and improves to no risk at the one week follow-up. If the participant maintains a passive risk at weekly follow-up for three follow-ups, the RA asks the participant to contact the RA for the 4th follow-up to confirm participant ability to proactively manage their own care. Follow-up ends when the participant successfully contacts the RA for follow-up.

**Active-low follow-up (SF 1, SF 3).** If the participant starts with active-low risk, follow-up is conducted weekly until the participant improves to no suicidal risk (in which case follow-up ends) or the participant improves to passive suicidal risk. If the latter, follow-up is completed according to the passive risk schedule. If the participant remains at passive or active-low risk for 3 follow-ups, the RA requests that the participant initiates the 4th follow-up contact. If the participant successfully contacts the RA for the 4th weekly follow-up, follow-up can end.

**Active-moderate follow-up (SF 1, SF 4).** Follow-up is conducted every 3 days for participants beginning with active-moderate risk. If the participant improves to active-low or passive risk, follow-up transitions to weekly follow-up according to the above guidelines. Follow-up can end if the participant improves to no suicidal risk at two consecutive weekly follow-ups. Follow-up ends without asking the participant to contact the RA for the last follow-up if the participant can confirm their health professional's contact information and that they will contact their mental health professional if thoughts of suicide return. The RA contacts the study clinical coordinator for further direction if the participant maintains active-moderate risk after 3 contacts.

**Active high and active-emergent follow-up (SF 1, SF 5).** If the participant begins with active-high or active-emergent risk, follow-up is conducted every day until the participant improves to active-moderate, active-low, or passive risk (in which case follow-up transitions to the follow-up for the participant's new risk category). If the participant has no suicidal risk at two consecutive weekly follow-ups, follow up can end. The RA is instructed to contact the study coordinator if the participant continues to have active-high or active emergent suicidal risk after 3 contacts.

In all risk levels, the RA instructs the participant to confirm that the participant has mental health professional contact information and will contact the health professional if thoughts of self-harm return. In all cases, the RA fills out the protocol form, including Section 6: Summary of Follow-up Contacts.

## Study analysis and statistics

We describe the feasibility of protocol implementation and discuss results of the implementation of our telephone protocol when participants screened positive for suicide risk during baseline, 3-, 6-, and 12-month surveys during the first 6 months of telephone protocol implementation (March 25, 2020-September 25, 2020). We use frequency distributions to highlight screened population characteristics, including demographic characteristics. Feasibility was measured as the proportion of completed follow-ups out of total number of follow-ups due as

indicated by the safety response protocol. We report the number of participants screened, the number of follow-ups completed, and the number of suicide risk cases resolved during telephone protocol use. We apply descriptive analyses to outline protocol implementation, including SRA screening and frequency of positive results. All statistical analyses were conducted using STATA IC (Version 16) software [43].

## Results

A total of n = 602 follow-up interviews among n = 738 eligible participants were conducted during the study period. The majority of participants were females aged 50 years or older who were married with children (Table 1). The majority (81%) of participants reported mobile phone access at baseline, with 2% having access to a landline at baseline.

### Feasiblity of protocol implementation and challenges

Study RAs used the protocol to successfully identify suicide risk level of all participants who screened positive for SI, with 100% of cases receiving the SRA being resolved. A total of 602 SI screenings took place in the first six months of telephone protocol implementation (during which no new participants were enrolled because of COVID-19 restrictions) (Table 2). During this period, 13 (2%) participants received a SRA after a positive PHQ-9 screening, with follow-up plans varying based on participant risk-level. Three (<1%) participants recieved a SRA after other indication of SI to study staff. In total, 15 (2%) participants received a SRA for any reason. All patients who screened positive received a SRA.

Most participants had passive risk (n = 11, 73%), followed by active-high (n = 2, 13%), active-moderate (n = 1, 7%), and active-low (n = 1, 7%) risk. Most participants with positive screenings reported suicidal thoughts 1–7 days in the previous two weeks (Table 3). Nearly one third of patients who screened positive during telephone protocol implementation (n = 5, 33%) screened positive at least once in the 6 months prior to telephone protocol implementation. No participants screened positive more than one time during telephone protocol use. There were no suicide attempts or deaths during the telephone implementation period.

The telephone protocol successfully guided the monitoring of all patients with SI until resolution of SI risk, with a 100% resolution rate of SI during this period indicating that this was an effective strategy to monitoring SI virtually. The first case during the telephone protocol period required study staff and the clinical lead reminding the RA to implement the new protocol. Only one patient required multiple contact attempts for assessment by the RA during their protocol follow-up period and all other contact attempts were successful on the first try. One participant conducted follow-up calls using a clinic phone during clinical visits. The mean duration of follow-up during the telephone protocol period was 14 days (range: [0, 35]), with the mean number of follow-ups was 2.4 (range: [0, 5]).

Researchers identified several challenges to telephone protocol implementation. Telephone visits were occasionally limited by cell phone access due to short, fixed-term phone plans and intermittent network problems affecting audio quality. Prepaid phones were able to receive calls from the research team and clinicians, but participants who had exhausted their prepaid minutes were unable to call research team members and clinicians until a new phone plan was purchased. These phone plans also resulted in some participants changing telephone numbers and access daily, risking delay of scheduled follow-up contacts or precluding follow-up contacts entirely. The use of a contact person's phone for follow-up risked delaying completion of SI assessments when the contact person was unwilling or unable to give the phone to the participant.

**Table 1. Baseline characteristics of SHARP participants*.**

|  |  | N (%) |
|---|---|---|
| **Age Group** |  | 738 |
|  | 18–29 | 22 (2.98) |
|  | 30–39 | 84 (11.38) |
|  | 40–49 | 196 (26.56) |
|  | 50+ | 436 (59.08) |
|  | Missing | 0 (0.00) |
| **Gender** | Female | 581 (78.73) |
|  | Male | 157 (21.27) |
|  | Missing | 0 (0.00) |
| **Employment** | Employed | 700 (94.85) |
|  | Unemployed | 34 (4.61) |
|  | Missing | 4 (0.54) |
| **Marital status** | Married | 489 (66.26) |
|  | Separated | 68 (9.21) |
|  | Divorced | 44 (5.96) |
|  | Widowed | 120 (16.26) |
|  | Cohabitating with partner | 3 (0.41) |
|  | Never married | 13 (1.76) |
|  | Missing | 1 (0.14) |
| **Parity**** | 0–4 | 192 (33.05) |
|  | 5–6 | 150 (25.82) |
|  | 7–8 | 142 (24.44) |
|  | 9–14 | 95 (16.35) |
|  | Missing | 2 (0.34) |
| **Phone access** | Mobile phone | 599 (81.17) |
|  | Non-mobile phone | 15 (2.03) |
| **SRA screening in 6 months prior to telephone protocol implementation**** | Yes | 5 (33.33) |
|  | No | 10 (66.67) |

*Of all participants eligible for current analysis (completed 0, 3, 6, and/or 12 month interviews and had a baseline PHQ-9 score of at least 5)

**n children, of female participants only

***Of those who screened positive in during the telephone protocol implementation period

RAs reported being unsure if the contact person was hiding information about harmful items in the home. Additionally, participants could not always receive the needed services when referred to the nearest health facility, forcing them to weigh the costs associated with traveling long distances to district health facilities to access mental health care. Transport

**Table 2. Frequency of factors related to Suicide Risk Assessment (SRA).**

| | | SRA telephone protocol period* |
|---|---|---|
| **Identification of suicidal ideation (SI)** | | 15 |
| | Positive PHQ-9 Question 9 | 12 (1.99) |
| | Otherwise reported SI | 3 (0.50) |
| | Missing | 0 (0.00) |
| **Self-harm or suicidal thoughts in the last 2 weeks** *among those who screened positive on PHQ-9 Question 9* | | 12 |
| | 1–7 days | 8 (66.67) |
| | 8–12 days | 1 (8.33) |
| | 13–14 days | 3 (25.00) |
| | Missing | 0 (0.00) |
| **Suicide risk level among those who screened positive on PHQ-9 Question 9 or otherwise reported SI** | | 15 |
| | Passive** | 11 (73.33) |
| | Active-Low** | 1 (6.67) |
| | Active-Moderate | 1 (6.67) |
| | Active-High/Active-Acute | 2 (13.33) |
| **Number of protocol follow-ups** | | |
| | Mean | 2.40 |
| | Median | 2.00 |
| | Min | 1 |
| | Max | 5 |
| **Duration of protocol follow-up (days)** | | |
| | Mean | 13.86 |
| | Median | 13.00 |
| | Min | 0 |
| | Max | 35 |

*SRA via telephone SRA analysis conducted for 6 months of telephone assessments after the start of the SARS-CoV-2 pandemic between March 25, 2020-September 24, 2020. Includes participants screened for SI during this study period.

**In-person assessment appeared to be more sensitive to active-low vs. passive risk in comparison to telephone assessments

***One contact (day of positive screening) only

problems from the home to the referred health facilities were a challenge to the referral process due to costs associated with travel. RAs reported that poor support services received by referred patients negatively affected trust between RAs, patients, and health providers in some cases.

## Case review

The study team generated adverse event reports weekly. One research coordinator and a psychiatrist reviewed all SI cases weekly. The coordinator asked each RA on a weekly basis if the SI protocol had been implemented in the last week. If so, the RA was instructed to follow up with the research coordinator individually. Noting that the protocol form itself is not submitted to the research team, but functions as a guide followed during telephone assessments, the RAs submitted a narrative of the telephone protocol event to the research team to review.

The research team met weekly to discuss fidelity to the protocol in each case, and one team member followed-up with RAs individually via WhatsApp messaging when more information on a case is needed or discussion about improvements to protocol fidelity was indicated [44].

**Table 3. PHQ-9 Question 9 participant responses among participants screened during the suicidality protocol implementation period.**

| | | Telephone PHQ-9 Q9 N (%) | Depressive severity | N (%) |
|---|---|---|---|---|
| **Baseline** | | 0* | | 0* |
| | 0 days | 0 (0.00) | No depression | 0 (0.00) |
| | 1–7 days | 0 (0.00) | Mild | 0 (0.00) |
| | 8–12 days | 0 (0.00) | Moderate | 0 (0.00) |
| | 13–14 days | 0 (0.00) | Moderately severe | 0 (0.00) |
| | Missing | 0 (0.00) | Severe | 0 (0.00) |
| | | | Missing | 0 (0.00) |
| **3 month follow-up** | | 83 | | 83 |
| | 0 days | 80 (96.39) | No depression | 46 (55.42) |
| | 1–7 days | 2 (2.41) | Mild | 33 (39.76) |
| | 8–12 days | 1 (1.20) | Moderate | 4 (4.82) |
| | 13–14 days | 0 (0.00) | Moderately severe | 0 (0.00) |
| | Missing | 0 (0.00) | Severe | 0 (0.00) |
| | | | Missing | 0 (0.00) |
| **6 month follow-up** | | 234 | | 234 |
| | 0 days | 231 (98.72) | No depression | 133 (56.84) |
| | 1–7 days | 3 (1.28) | Mild | 73 (31.20) |
| | 8–12 days | 0 (0.00) | Moderate | 21 (8.97) |
| | 13–14 days | 0 (0.00) | Moderately severe | 5 (2.14) |
| | Missing | 0 (0.00) | Severe | 0 (0.00) |
| | | | Missing | 2 (0.85) |
| **12 month follow-up** | | 285 | | 285 |
| | 0 days | 279 (97.89) | No depression | 168 (58.95) |
| | 1–7 days | 3 (1.05) | Mild | 93 (32.63) |
| | 8–12 days | 0 (0.00) | Moderate | 13 (4.56) |
| | 13–14 days | 3 (1.05) | Moderately severe | 5 (1.75) |
| | Missing | 0 (0.00) | Severe | 4 (1.40) |
| | | | Missing | 2 (0.70) |

*No new patients were enrolled during the study period due to the pandemic

**n = 2 participants are missing overall PHQ-9 score

The research team closed each protocol case when the weekly review team agreed that all protocol components had been completed with fidelity and that suicide risk was no longer present. In these meetings, the team also discussed if protocol modification was needed to address challenges being faced by the research team (e.g., how to modify the protocol in the event of chronic suicidal ideation).

## Protocol implementation example

The following case study demonstrates the application of the telephone-based SRA protocol.

Day 0 –Study RA reported that ODK directed them to complete an SRA with Participant A during a routine, phone-based data collection interview. After completing the SRA, the result was Passive SI. The RA followed the 'Passive SI' protocol steps and determined that the participant did not have access to items they could use to harm themselves, established contact with the participant's spouse (should the RA be unable to reach the participant), and provided mental health referral information should the participant's suicidal thoughts worsen. The

participant mentioned to the RA that they did not feel like they were currently a danger to themselves or others and said they would follow-up with the referral.

Day 7 –Study RA contacted the participant by phone one week later with fidelity, consistent with the protocol's Passive SI guidelines. During this call the participant endorsed that they no longer had thoughts of harming themself by answering "no" to PHQ-9 question 9.

Day 9 –Study clinical lead met with study coordinator to discuss the case. After noting the RA's fidelity to the protocol and Participant A's improvement at follow-up, the case was closed resulting in no further follow-up.

One key point illustrated by this case study is that, although the full protocol is very detailed, its implementation typically was quite efficient in practice. In fact, the protocol was not needed for 98% of study contacts, and when it was needed, safety concerns were appropriately handled with a small amount of follow-up contact. Thus, while a detailed protocol that anticipated all possible eventualities and backup processes was essential, the actual added effort for the research team was quite modest.

## Discussion

Adding to the baseline challenges of assessing and managing suicidal behavior, the SARS-CoV-2 pandemic has disrupted and limited mental health care by precluding many face-to-face assessments, which has been especially problematic for the assessment and management of suicidality [25, 28]. Currently, 93% of countries are experiencing disruptions to mental healthcare and 24% of countries reported a disruption in suicide prevention programs due to the pandemic [45]. Increased suicidal risk has previously been associated with epidemics, with possibility of increased risk of suicide during the current SARS-CoV-2 pandemic [29–32]. This paper reports on a feasible strategy to monitor suicide risk by phone when in-person assessments are not possible.

The rapid adaptation of SHARP's in-person SRAs for telephone use during the SARS-CoV-2 pandemic allowed us to continue monitoring SI in an ongoing RCT in Malawi during the pandemic. The possibility of increasing suicide risk during the pandemic is concerning, and adapting and implementing a feasible, virtual protocol that addresses study participant suicide risk events and appropriately refers participants to care is critical [45]. Similar SRA protocols have been adapted for use in RCTs, but few have been developed in non-hospital, emergency, and time-sensitive contexts in which mental health care is severely disrupted [46–49].

Overall, transition from in-person to telephone-based SRAs was feasible and functioned well for patients and research team members. Providing a structured protocol is critical to providing research team members with the support and guidance needed to assess participant suicide risk [46]. The protocol identified participants at risk of suicide and referred participants to clinical settings when indicated. As well, the protocol provided research team members with feasible, structured, and actionable responses to participant SI.

Very few participants had active-low or active-high suicide risk during the telephone protocol period, contrary to what we anticipated based on current literature, subject matter expertise, and prior in-person SRAs [50, 51]. With the addition of the "Thoughts of hurting yourself right now" response option, the protocol treats acute-high and active-emergent cases as equally serious with the same follow-up intensities. As well, the "Might hurt yourself before seeing your doctor again" response option may be less meaningful during pandemic times if clinical visits are postponed. Future directions include validation studies to address these questions.

Importantly, one third of participants who had a positive SRA screening during the telephone protocol implementation had a positive SRA screening in the 6 months prior to telephone

protocol implementation, a reminder of the importance of feasible and well-designed SRA follow-up schedules [52, 53].

Challenges cited by research team members include inability to predict certain challenges faced during the pandemic (e.g., changes in governmental and institutional policies about quarantine, travel, and other factors affecting participants), network and audio difficulties, and risk of participant mobile phone access delaying or precluding follow-up visits [54–56]. As some participants had phone access through close contacts reported in SRA Sections 2 and 3, it was criticial to collect this information during assessments and participant follow-up was occasionally limited by access to the contact person's phone. Phone minute-related challenges limited participant ability to initiate calls with clinicians and research team members, despite being able to receive calls from the research team. RAs identified challenges to referring participants to health centers, as not all centers could provide the needed services. Despite inability to predict all possible challenges, the research team created a protocol and multi-layered review process robust enough to meet challenges and circumstances faced by participants and research staff through iterative adjustment and adaptation.

This study includes a small sample of adult participants, limiting generalizability. Lack of availability of qualitative fidelity data for all events precluded formal analysis of protocol fidelity. However, all RA protocol-related actions and responses were monitored by the research team through a weekly, iterative review process and real-time correspondence via WhatsApp. Finally, it is possible that participant responses to questions about suicidality differ when asked about suicidality by phone versus in person, and possibly undreport symptoms [19]. But, prior SI scales have performed well under rapid, self-reporting conditions conditions [18, 22, 49]. The protocol performed conservatively, to error on classifying patients as having a higher severity status. As well, in-depth, weekly case-reviews of all suicide risk events and RA responses by the research team ensured that the protocol was implemented accurately and that participants received appropriate and timely follow-up and care when clinically indicated.

Research implications include the need to assess the protocol's sensitivity, specificity, positive predictive value (PPV), and negative predicative value (NPV) [57, 58]. Implications also include identifying and including variables for implicit suicidal thoughts to more accurately identify active-low compared to active-moderate and active-high participants, as this may result in different clinical action indication [58–60].

This protocol can serve as a model for adaptation of adult SI protocols for telephone or virtual use in times of pandemic or not. Further, this protocol can function as a model for use in other contexts, particularly in adult populations in LMICs.

## Conclusions

Mental health care barriers magnified by the SARS-CoV-2 pandemic highlight the increasing need to address mental healthcare through technology. In order to conduct mental health research or health care remotely, researchers and clinicians must have access to standardized, reliable, and accurate suicide risk assessment protocols. The telephone suicide risk assessment protocol developed to assess SHARP participant suicide risk during the SARS-CoV-2 pandemic can serve as a model for development of virtual suicide risk assessment protocols, in times of pandemic or not.

## Supporting information

**S1 File. SHARP study safety response protocol for phone interviews.**
(PDF)

**S2 File. Decision tree for SHARP safety response protocol for phone interviews.**
(PDF)

**S3 File. SHARP safety response protocol for phone interviews decision tree for passive and active-low suicide risk follow-up.**
(PDF)

**S4 File. SHARP safety response protocol for phone interviews decision tree for moderate suicide risk follow-up.**
(PDF)

**S5 File. SHARP safety response protocol for phone interviews decision tree for active high and active emergent suicide risk follow-up.**
(PDF)

**S6 File. Inclusivity in global research.**
(PDF)

## Author Contributions

**Conceptualization:** Kelsey R. Landrum.

**Data curation:** Kelsey R. Landrum, Christopher F. Akiba, Brian W. Pence, Harriet Akello, Hamis Chikalimba, Josée M. Dussault, Mina C. Hosseinipour, Kingsley Kanzoole, Kazione Kulisewa, Jullita Kenala Malava, Michael Udedi, Chifundo C. Zimba, Bradley N. Gaynes.

**Formal analysis:** Kelsey R. Landrum.

**Funding acquisition:** Kelsey R. Landrum, Josée M. Dussault.

**Investigation:** Christopher F. Akiba, Brian W. Pence, Harriet Akello, Hamis Chikalimba, Josée M. Dussault, Mina C. Hosseinipour, Kingsley Kanzoole, Kazione Kulisewa, Jullita Kenala Malava, Michael Udedi, Chifundo C. Zimba, Bradley N. Gaynes.

**Methodology:** Kelsey R. Landrum, Christopher F. Akiba, Brian W. Pence, Josée M. Dussault, Bradley N. Gaynes.

**Project administration:** Kelsey R. Landrum.

**Supervision:** Brian W. Pence, Bradley N. Gaynes.

**Visualization:** Kelsey R. Landrum.

**Writing – original draft:** Kelsey R. Landrum, Christopher F. Akiba, Bradley N. Gaynes.

**Writing – review & editing:** Kelsey R. Landrum, Christopher F. Akiba, Brian W. Pence, Harriet Akello, Hamis Chikalimba, Josée M. Dussault, Mina C. Hosseinipour, Kingsley Kanzoole, Kazione Kulisewa, Jullita Kenala Malava, Michael Udedi, Chifundo C. Zimba, Bradley N. Gaynes.

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
