## [Decision Letter · Decision Letter 0]

30 Sep 2022

PONE-D-22-16562Assessing suicidality during the SARS-CoV-2 pandemic: Lessons learned from adaptation and implementation of a telephone-based suicide risk assessment and response protocol in MalawiPLOS ONE

Dear Dr. Landrum,

Thank you for submitting your manuscript to PLOS ONE. After careful consideration, we feel that it has merit but does not fully meet PLOS ONE’s publication criteria as it currently stands. Therefore, we invite you to submit a revised version of the manuscript that addresses the points raised during the review process.

I'm sorry for the significant time it has taken to review this manuscript. Kindly address the comments provided by the reviewers.==============================

We look forward to receiving your revised manuscript.

Kind regards,

Ching Sin Siau

Academic Editor

PLOS ONE

Journal Requirements:

3. We note that you have stated that you will provide repository information for your data at acceptance. Should your manuscript be accepted for publication, we will hold it until you provide the relevant accession numbers or DOIs necessary to access your data. If you wish to make changes to your Data Availability statement, please describe these changes in your cover letter and we will update your Data Availability statement to reflect the information you provide

Reviewers' comments:

Reviewer's Responses to Questions

**Comments to the Author**

1. Is the manuscript technically sound, and do the data support the conclusions?

Reviewer #1: Yes

Reviewer #2: Yes

2. Has the statistical analysis been performed appropriately and rigorously? 

Reviewer #1: Yes

Reviewer #2: Yes

3. Have the authors made all data underlying the findings in their manuscript fully available?

Reviewer #1: No

Reviewer #2: Yes

4. Is the manuscript presented in an intelligible fashion and written in standard English?

Reviewer #1: Yes

Reviewer #2: Yes

5. Review Comments to the Author

Reviewer #1: The SARS-CoV-2 pandemic led to the rapid transition of many research studies from in-person to telephone follow-up. It calls on efforts to develop and adapt protocols to fit this change. This paper describes the feasibility of development and implementation of a suicidality assessment protocol. It assessed its feasibility for assessing suicidality over telephone among patients who screen positive during the SRA, described the information collected and categorized, and described the primary challenges identified and lessons. This research belongs to a much needed type of studies to solve the urgent issues we face. The protocol / study is well described and the changeless and lessons are well discussed.

I just have a minor comment suggesting clarifying some numbers of the results—there were 689 patients meeting the criteria, but there are only 604 SI screens took place? There were 13 PHQ-9 positive, but only 12 received SRA?

In Abstract, the abbreviate SI was first used without the full name spelled (line 59), and then was done on line 61.

Reviewer #2: it's an interesting and topical important study to be conducted and applied to other setting in the future. Overall, a good manuscript. Few questions seeking for clarification:

1. Are there any expert validation being done? Could elaborate further in line 199?

2. Who provided the training- further elaborate in line 205

3. Would the RA has specific academic or research background requirement? Kindly also include that, if any.

4. Was English the communication and main & only language used in the validation process with the participants?

5. Line 169-175; there was a mixed of person interviews+ telephone being the data collection method. Would this present as a form of limitation? If yes- perhaps to include as limitation? If not- how do you mitigate the effect?

6. PLOS authors have the option to publish the peer review history of their article (what does this mean?). If published, this will include your full peer review and any attached files.

Reviewer #1: No

Reviewer #2: No

---

## [Author Response · Author response to Decision Letter 0]

8 Dec 2022

November 11, 2022

PLOS One 

RE: Manuscript Title: “Assessing suicidality during the SARS-CoV-2 pandemic: Lessons learned from adaptation and implementation of a telephone-based suicide risk assessment and response protocol in Malawi”, Manuscript Number: PONE-D-22-16562R1

Dear Reviewers 1 and 2,

Thank you very much for your time and thoughtful comments on our manuscript submitted to PLOS ONE, titled “Assessing suicidality during the SARS-CoV-2 pandemic: Lessons learned from adaptation and implementation of a telephone-based suicide risk assessment and response protocol in Malawi”. These comments are helpful in revising our manuscript. We have responded to your comments in line below, as well as in the manuscript through tracked changes. We hope that these changes adequately address your questions and comments.

Sincerely,

Kelsey R. Landrum

Reviewer #1: The SARS-CoV-2 pandemic led to the rapid transition of many research studies from in-person to telephone follow-up. It calls on efforts to develop and adapt protocols to fit this change. This paper describes the feasibility of development and implementation of a suicidality assessment protocol. It assessed its feasibility for assessing suicidality over telephone among patients who screen positive during the SRA, described the information collected and categorized, and described the primary challenges identified and lessons. This research belongs to a much needed type of studies to solve the urgent issues we face. The protocol / study is well described and the changeless and lessons are well discussed.

I just have a minor comment suggesting clarifying some numbers of the results—there were 689 patients meeting the criteria, but there are only 604 SI screens took place? There were 13 PHQ-9 positive, but only 12 received SRA?

Thank you for this comment and clarification. We have updated the data since this manuscript was submitted for review. The new manuscript addresses this discrepancy (as one participant had been receiving SRAs due to prevalent SI identified earlier in the parent study and not on a PHQ9 screening during the time period for the current analysis). The updated manuscript clarifies this and contains up-to-date data. 

In Abstract, the abbreviate SI was first used without the full name spelled (line 59), and then was done on line 61.

Thank you for this comment. We had ensured that suicidal ideation is first spelled out, then abbreviated appropriately.

Reviewer #2: it's an interesting and topical important study to be conducted and applied to other setting in the future. Overall, a good manuscript. Few questions seeking for clarification:

1. Are there any expert validation being done? Could elaborate further in line 199?

Thank you for this question. Given the immediate need for a suicidality protocol during the pandemic, the guide was developed by subject matter experts and iteratively adapted. We have clarified this in the ‘Protocol development section’.

2. Who provided the training- further elaborate in line 205

Research assistance were trained by subject matter experts in clinical psychiatry, health behavior, and epidemiology. We have clarified this in lines 208-210.

3. Would the RA has specific academic or research background requirement? Kindly also include that, if any.

All RA’s had a minimum of a high school education level and were trained in SHARP study procedures and suicidality protocol procedures. We have clarified this in lines 208-209.

4. Was English the communication and main & only language used in the validation process with the participants?

Thank you for this clarification. The protocol was conducted in Chichewa or Chitumbuka, with responses translated into English (lines 212-215). We have ensured this is clarified in the ‘Protocol development’ section.

5. Line 169-175; there was a mixed of person interviews+ telephone being the data collection method. Would this present as a form of limitation? If yes- perhaps to include as limitation? If not- how do you mitigate the effect?

Thank you for this comment. We agree that this could be a limitation in that participant responses to interview questions may differ by mode of interview (in person versus via phone) and have added clarification about this potential form of bias in our discussion section. The study team, including a psychiatry study team member, conducted weekly reviews of all SRAs and patient follow-ups to ensure timely receipt of clinical care for all patients being followed up for SI. As we note, the phone protocol may be able to less accurately predict active-low compared to active-moderate suicide risk compared to the in-person assessment. In other words, as discussed in the discussion, the protocol tends to perform conservatively and classify patients as having a higher severity level when it is unclear if patients are a true low severity level. The literature has not measured the extent of this limitation, as most SRAs are conducted in person by clinical staff, but we have added more discussion of this potential limitation.1–7 

 

REFERENCES

1. Harbauer, G., Ring, M., Schuetz, C., Andreae, A. & Haas, S. Suicidality assessment with PRISM-S—Simple, fast, and visual: A brief nonverbal method to assess suicidality in adolescent and adult patients. Crisis: The Journal of Crisis Intervention and Suicide Prevention 34, 131–136 (2013).

2. Viguera, A. C. et al. Comparison of Electronic Screening for Suicidal Risk With the Patient Health Questionnaire Item 9 and the Columbia Suicide Severity Rating Scale in an Outpatient Psychiatric Clinic. Psychosomatics 56, 460–469 (2015).

3. Forkmann, T. et al. Assessing suicidality in real time: A psychometric evaluation of self-report items for the assessment of suicidal ideation and its proximal risk factors using ecological momentary assessments. Journal of Abnormal Psychology 127, 758–769 (2018).

4. Ellis, T. E., Rufino, K. A. & Allen, J. G. A controlled comparison trial of the Collaborative Assessment and Management of Suicidality (CAMS) in an inpatient setting: Outcomes at discharge and six-month follow-up. Psychiatry Research 249, 252–260 (2017).

5. Exbrayat, S. et al. Effect of telephone follow-up on repeated suicide attempt in patients discharged from an emergency psychiatry department: a controlled study. BMC Psychiatry 17, (2017).

6. Garza, N. D. L. et al. The Concise Health Risk Tracking Self-Report (CHRT-SR) assessment of suicidality in depressed outpatients: A psychometric evaluation. Depression and Anxiety 36, 313–320 (2019).

7. Herbeck Belnap, B. et al. Electronic protocol for suicide risk management in research participants. Journal of Psychosomatic Research 78, 340–345 (2015).

---

## [Decision Letter · Decision Letter 1]

31 Jan 2023

Assessing suicidality during the SARS-CoV-2 pandemic: Lessons learned from adaptation and implementation of a telephone-based suicide risk assessment and response protocol in Malawi

PONE-D-22-16562R1

Dear Dr. Landrum,

We’re pleased to inform you that your manuscript has been judged scientifically suitable for publication and will be formally accepted for publication once it meets all outstanding technical requirements.

Kind regards,

Ching Sin Siau

Academic Editor

PLOS ONE

Additional Editor Comments (optional):

Reviewers' comments:

Reviewer's Responses to Questions

**Comments to the Author**

1. If the authors have adequately addressed your comments raised in a previous round of review and you feel that this manuscript is now acceptable for publication, you may indicate that here to bypass the “Comments to the Author” section, enter your conflict of interest statement in the “Confidential to Editor” section, and submit your "Accept" recommendation.

Reviewer #1: All comments have been addressed

Reviewer #2: All comments have been addressed

2. Is the manuscript technically sound, and do the data support the conclusions?

Reviewer #1: Yes

Reviewer #2: Yes

3. Has the statistical analysis been performed appropriately and rigorously? 

Reviewer #1: N/A

Reviewer #2: Yes

4. Have the authors made all data underlying the findings in their manuscript fully available?

Reviewer #1: No

Reviewer #2: Yes

5. Is the manuscript presented in an intelligible fashion and written in standard English?

Reviewer #1: Yes

Reviewer #2: Yes

6. Review Comments to the Author

Reviewer #1: My comments were addressed. There is no further critique.

My comments were addressed. There is no further critique.

Reviewer #2: The authors have addressed all comments in the revised manuscript. Sounds justification has been provided.

It's good to go.

7. PLOS authors have the option to publish the peer review history of their article (what does this mean?). If published, this will include your full peer review and any attached files.

Reviewer #1: No

Reviewer #2: **Yes: **OOI PEI BOON

---

## [Editor Report · Acceptance letter]

8 Mar 2023

PONE-D-22-16562R1 

Assessing suicidality during the SARS-CoV-2 pandemic: Lessons learned from adaptation and implementation of a telephone-based suicide risk assessment and response protocol in Malawi 

Dear Dr. Landrum:

I'm pleased to inform you that your manuscript has been deemed suitable for publication in PLOS ONE. Congratulations! Your manuscript is now with our production department. 

Kind regards, 

on behalf of

Dr. Ching Sin Siau 

Academic Editor

PLOS ONE